# Oral Diseases and Quality of Life between Obese and Normal Weight Adolescents: A Two-Year Observational Study

**DOI:** 10.3390/children8060435

**Published:** 2021-05-22

**Authors:** Tengku Nurfarhana Nadirah Tengku H, Wei Ying Peh, Lily Azura Shoaib, Nor Adinar Baharuddin, Rathna Devi Vaithilingam, Roslan Saub

**Affiliations:** 1Department of Paediatric Dentistry and Orthodontics, Faculty of Dentistry, University Malaya, Kuala Lumpur 50350, Malaysia; lilyazura@um.edu.my; 2Klinik Pergigian Cheras, Jalan Yaacob Latiff, Cheras, Kuala Lumpur 56000, Malaysia; wying18@hotmail.com; 3Department of Restorative Dentistry, Faculty of Dentistry, University Malaya, Kuala Lumpur 50350, Malaysia; noradinar@um.edu.my (N.A.B.); rathna@um.edu.my (R.D.V.); 4Department of Community Oral Health & Clinical Prevention, Faculty of Dentistry, University Malaya, Kuala Lumpur 50350, Malaysia; roslans@um.edu.my

**Keywords:** obesity, oral disease, oral health related quality of life (OHRQoL)

## Abstract

This study aimed to investigate the association between oral disease burden and oral health related quality of life (OHRQoL) among overweight/obese (OW/OB) and normal weight (NW) Malaysian adolescents. A total of 397 adolescents were involved in the two-year prospective observational cohort study. OHRQOL was measured through a self-administered questionnaire containing the short version of the Malaysian Oral Health Impact Profile (OHIP[M]). Body mass index (BMI) was used for anthropometric measurement. Whilst, decayed, missing, and filled teeth (DMFT) index, Significant Caries Index (SiC), simplified basic periodontal examination (S-BPE), and gingival bleeding index (GBI) were used for clinical assessment tools. Higher dental caries prevalence was observed in the NW group while higher SiC was reported in the OW/OB group. Regardless of the obesity status, the prevalence of gingivitis (BPE code 1 and 2) was high in this study. A reduction of GBI prevalence was observed in the two-year follow-up results with an increased prevalence of OHRQoL impact in the OW/OB group compared to the NW group (*p* > 0.05). The findings from this study suggested that obesity status did not have influence over the burden of oral diseases and OHRQoL. It offers insights referring to the changes in adolescents’ oral diseases burden and OHRQoL.

## 1. Introduction

Overweight and obesity are defined as abnormal or excessive fat accumulation in the body, which can impact one’s health [1]. The current global trend showed that obesity has become a serious public concern regardless of a country’s development status [2]. Globally, over 340 million children and adolescents aged five to nineteen were either overweight or obese in 2016 [1]. The Malaysian National and Health Morbidity Survey (NHMS) reported that 29.8 percent of adolescents (aged 13–17 years old) were overweight/obese (OW/OB) in Malaysia [3]. This is a situation that we cannot afford to neglect as 40 percent of the current Malaysian demographic profile range from 0 to 24 years of age.

Research showed obesity is likely to progress from adolescence to adulthood [4]. This is rather a worrying phenomenon as it will potentially impair general and oral health in both adolescence and adulthood. To date, research on oral health has been primarily conducted on OW/OB adults, with less attention given to adolescents. Different studies are required in adolescence, as they are incomparable with existing studies that have been carried out in adults. Unlike with adults, the puberty hormonal interaction alters the body composition and regional distribution of body fat in adolescents.

The most common oral health problems are dental caries, while periodontal disease is known as the eleventh most prevalent condition globally [1]. In general, obesity, dental caries, and periodontal disease share common risk factors, namely: sugary diet, dental plaque, socioeconomic status, and behavioral and genetic issues [5,6,7]. Although, there is scarce evidence of a long-term relationship among OW/OB, poor oral health status, and oral health-related quality of life (OHRQoL), some studies suggest positive associations [5,8]. Compromised health-related quality of life (HRQoL) associated with increased age and gender in OW/OB adolescents was also reported [9].

It is essential to further scrutinize the relationship between obese adolescence and changes in oral health status through longitudinal studies as the establishment between obesity and oral health will lead to a huge global economic impact due to the rising costs attributed to obesity and oral diseases [10]. This, in a way, will prompt health policymakers in prioritizing action and evaluating preventive measures for the OW/OB adolescence oral health. Thus, this longitudinal study aimed to investigate the association of OW/OB in adolescence with oral disease burden (dental caries and periodontal disease) and OHRQoL over a two-year period.

## 2. Materials and Methods

### 2.1. Study Design and Study Population

This is a prospective observational cohort study on overweight/obese and normal weight schoolchildren residing in the districts of Klang Valley, Selangor. A stratified multiple-stage random sampling technique was employed to recruit secondary school adolescents aged fourteen for the baseline study in the year 2015. The follow-up study was conducted in the year 2017 within the same group of participants at the age of sixteen. Malaysian citizens who were fit and healthy, able to understand and communicate well in Malay language, and attending multi-racial co-educational daily public school were recruited. Participants undergoing orthodontics treatment, or had underweight BMI status, or had a change of BMI status during the two-year follow-up were excluded.

A total of 202 OW/OB participants were cross matched by age and gender with 195 NW participants who fulfilled the inclusion criteria. Anthropometric measurement and clinical examination including caries experience and periodontal status were performed following the completion of self-administered questionnaires consisting of sociodemographic data, oral health related behaviors, and the short version of the Malaysian Oral Health Impact Profile (OHIP[M]).

### 2.2. Ethics, Consents, and Permission

Ethical approval (DF CD1706/0031(U)) and permission to conduct the study at public school was obtained from the Faculty of Dentistry Medical Ethics Committee, University of Malaya, Ministry of Education Malaysia, Selangor State Education Department, and authorities of the selected secondary schools. Written informed consent from the parents and caregivers of the participants were obtained at both baseline and follow-up studies. Participations for both studies were on a voluntary basis.

### 2.3. Anthropometric Measurement

The 2007 WHO reference was used to determine each participant’s body mass index (BMI) status [1].

The participants were instructed to only wear standard school uniforms and socks upon standing on the weight scale platform during body height and body weight measurement. The body weight was measured to the nearest 0.5 kg and height to the nearest 0.1 cm using a calibrated digital scale with body mass index (BMI) function and integrated stadiometer (Tanita, IL, USA), where routine calibrations were conducted based on manufacturer recommendations.

### 2.4. Clinical Examinations

#### 2.4.1. Caries Experiences

The caries experienced was recorded following the WHO criteria [11], where the DMFT score was calculated by adding the number of decayed (D), missing (M), and filled (F) permanent teeth. The Significant Caries Index (SiC) was used to identify individuals with the highest caries values in the population. SiC was determined by selecting one-third of the samples with highest DMFT scores and the mean DMFT for this subgroup was calculated [12].

#### 2.4.2. Periodontal Assessments

The Simplified Basic Periodontal Examination (S-BPE) by the British Society of Periodontology (BSP) and British Society of Pediatric Dentistry (BSPD) was used for the screening of periodontal health. The examination was performed on the following six index teeth, 16, 11, 26, 36, 31, and 46, using the WHO probe with the ball end tip. The BPE codes (Code 0 to Code 2) were used to determine each participant’s periodontal status. The Ainamo and Bay Gingival Bleeding Index (GBI) [13] was used as an adjunct to the BPE to determine each participant’s gingival health by gently probing on the orifice of the gingival crevice to the bottom of the pocket. Positive findings were recorded if bleeding occurred within 10 s.

#### 2.4.3. Training and Calibration

All participating dentists involved in this study were trained and calibrated to minimize inter-examiner variability in the data collection. The calibration sessions were carried out prior to the data collection at baseline and the follow-up study for caries diagnoses, periodontal and gingival examination to a single gold standard examiner of each field (pediatric dentist and periodontist). All examiners needed to achieve the minimum required kappa value of 0.75 prior to the conduct of the study. The kappa values of the examiners compared to the gold standard ranged from 0.75 to 0.80. As the standard operating procedure for this study, the intra-examiner’s reliability was conducted at each data collection session.

### 2.5. Questionnaires

A self-administered questionnaire was used to collect participants’ information on demographics, oral health related behaviors, and oral health related quality of life (OHRQoL) using the short form of the Malaysian version of Oral Health Impact Profile OHIP[M]-14 that contained 14 items in seven domains. The two parameters obtained from OHIP[M]-14 were prevalence and severity. Prevalence of impact is the percentage of respondents reporting one or more impacts of “fairly often” or “often”, whereas severity impact was the summation of response codes for the fourteen items. The content of the questionnaire was validated during the baseline study by two dental specialists from pediatric dentistry and dental public health. It was pre-tested for clarity, simplicity, sequencing, and ease of understanding instructions and questions. The internal consistency measured by Cronbach’s alpha was recorded at 0.75, which indicated acceptable reliability.

### 2.6. Statistical Analysis

All data collected were treated confidentially and verified by the examiner for its completeness prior to data entry and data analysis. The data were coded and recorded using IBM SPSS Version 26.0 (SPSS Inc., Chicago, IL, USA). The *p*-value was set at α = 0.05 with 95% confidence interval.

Differences between baselines and follow-up distribution of gender, race, mother level of education, and body mass index were compared between participants who successfully completed each phase of data collection through the Cohen effect size (r). The characteristics of participants were described using frequency distribution for categorical variables while continuous data were described using mean (M) and standard deviation (±SD).

The chi square statistic (X^2^) was used for comparison of oral hygiene practice and self-rated perception of health and oral health between obese and normal weight groups and the association was reported as statistically significance if the *p* value < 0.05.Comparison for mean DMFT, mean SiC, and severity of OHRQoL impact between obese and normal weight groups were calculated using the independent t-test. The chi square test was used to compare the prevalence of dental caries, periodontal disease, gingival bleeding, and impact of OHRQoL between the two groups.

## 3. Results

Out of 397 participants (NW and OW/OB) recruited at the baseline study (age fourteen), only 238 (60%) participants who fulfilled the inclusion criteria managed to be traced and agreed to participate in the follow-up study (age sixteen). The sociodemographic and anthropometric characteristics of the follow-up participants were comparable to the dropout participants (Cohen’s effect size ≤ 0.5) (Table 1). The reasons for participants’ dropout included change of schools, not willing to participate, being absent during data collection day, and not fulfilling the criterion. The final analysis of this study involved the total number of those who participated at both baseline and follow-up study.

### 3.1. Sociodemographic

The sociodemographic characteristics of participants in the baseline and follow-up study were almost similar across BMI groups. Participants for the follow-up study consisted of 146 males and 92 females with 60% of the participants from Malay ethnicity. It was found that the majority of the participants’ mothers came from lower educational backgrounds. The mean baseline BMI of the overweight/obese group was 28.5 kg/m^2^ and the normal weight group was 19.2 kg/m^2^. The mean BMI of OW/OB (29.6 kg/m^2^ ± 4.30) and NW (19.6 kg/m^2^ ± 1.67) increased at the follow-up study compared to the baseline study.

### 3.2. Oral Health Practices

Good tooth brushing habits were seen in more than 40% of the participants across all groups. The majority of the participants in the SiC group brushed their teeth twice daily by using fluoridated toothpaste and using mouth rinses (Table 2). However, a two-fold increment in the number of participants not using fluoridated toothpaste was seen in OW/OB and NW groups during the follow-up study.

Irrespective of BMI status, an increase in the number of participants who rated poor on perception of health and oral health was observed in the follow-up study. The same pattern was also observed among participants in the SiC group. In contrast, the percentages of participants visiting dentists within one-year remained low in both baseline and follow-up studies.

### 3.3. Caries and Periodontal Health

The mean (M ± SD) caries experiences among OW/OB and NW groups in the baseline were 1.80 ± 3.16, 1.84 ± 2.62, with a slight increment to 2.37 ± 3.30 and 2.47 ± 3.05, respectively at the two-year follow-up study (Table 3). Likewise, the number of teeth affected by carious lesions showed an increasing trend. High mean of caries experiences increased from 5.22 to 6.49 per participants in the OW/OB of SIC group at the baseline and follow-up study. In regards to the SiC participants, nearly 90% did not visit the dentist for the past one year at the time when the study was conducted.

The distribution of BPE scores are shown in Table 3 for OW/OB and NW of the total population and SIC group. The highest score observed in the study population was “Code 2”, which can be described as presentation of calculus or plaque retention factor with no periodontal pocket ≥3.5 mm detected. At the baseline study, 77% of participants were presented with “Code 2” (Table 3). During the follow-up study, it was observed that the BPE scores reduced more than 20% across all participants, including in the SIC group. Gingival bleeding can be observed in more than 90% of participants at baseline and follow-up study.

### 3.4. Oral Health Related Quality of Life

The oral health related quality of life among OB/OW, NW, and SIC groups at baseline and follow-up are reported in Table 4. The Oral Health Impact Profile 14 (OHIP 14) scores ranged from 0 to 56, with a high score indicating poor oral health related quality of life. The OW/OB participants in the SIC group presented with the highest mean score of 9.83 ± 6.68 at baseline and 8.18 ± 7.47 during the follow-up study (*p* > 0.05). In general, the prevalence of impact was low in all domains at baseline and the follow-up study. There were four most affected domains, with at least 10% of OW/OB groups involved. The “self-consciousness” and “felt tense” items in the psychological discomfort domain were the two most affected items among the SIC group at baseline. The prevalence increase between 4% and 10% was observed at the follow-up study in the same group. The social disability domain (been irritable with others) was also reported as the affected domain at 11.4%, followed by functional limitation (worsening in the sense of taste).

## 4. Discussion

The aim of this study was to present the results of a longitudinal study on the severity of dental caries and periodontal health among obese and normal weight Malaysian adolescents and the impact on oral health related quality of life. Despite the good response rate in the present study, the refusal to participate in the follow-up study may represent a source of selection bias. However, the effect size of participants that turned up and lost to follow-up was low (Cohen effect size < 0.5), not affecting the overall result of this study.

Severe obese individuals were reported with risk of developing dental caries, periodontal disease, and health morbidity [7,8,14]. However, it was not reflected in the findings of this study. It can be explained by the recruitment of obese participants in this study—it was not a true or severe obese population by definition (BMI > 25.9 kg/m^2^ for boys and BMI > 27.3 kg/m^2^ for girls at the age of 14 years old). Even though, most participants were overweight, it was important to start planning for an early prevention for one-third of the participants who presented with a high burden of oral diseases. It is evident that BMI provides an excellent indicator for obesity, but studies proved that waist circumference is the best index for predictor of obesity related health risks [14].

This study adopted the WHO growth reference, which was also employed in the National Physical Fitness Standard for Malaysia School Students (SEGAK) program. SEGAK was first implemented by the Malaysian Ministry of Education in 2008 for Malaysian students aged 10 to 17 years old [15]. It was designed to evaluate each student’s level of fitness and, at the same time, monitor their BMI. By adopting the same reference, it allowed comparison of results to be made if similar future studies were to be conducted in other states in Malaysia.

From the sociodemographic background related to maternal education, the majority were from lower education levels. The majority were reported to have at least attended secondary school. Therefore, it can be suggested that most of the participants came from low socioeconomic status backgrounds. Growing interest in studies related to low socioeconomic status and development of childhood obesity can be seen in few areas, particularly in mental health, impact on health-related quality of life, and financial hardship. These areas affect healthy lifestyle choices that are less accessible and indirectly promote calorie dense foods, lifestyles that reflect a lack of physical activity, poor oral hygiene practices, and less opportunities for self-growth and development, especially in education [16,17]. The number of participants who rated good on ‘perception of health and oral health’ also showed a reduction in the follow-up study compared to the baseline study across the BMI status. In contrast, the percentages of participants visiting dentists within one year remained low in both baseline and follow-up studies. This can be postulated from the statistically non-significant (*p* > 0.05) association between BMI status with the burden of oral diseases and impact on oral health related quality of life.

Good tooth brushing habits were also being observed with the majority of participants brushing their teeth twice daily and using fluoridated toothpaste. However, a two-fold increment in the number of participants not using fluoridated toothpaste was seen in both groups at the follow-up study. Providentially, the increment in the mean of caries was not significant. This might be due to the misconceptions by some participants and their parents, regarding the fluoride concentration levels in the toothpaste. Despite using non-fluoridated toothpaste, the participants still benefited (i.e., via a caries-preventive effect) through public water fluoridation, provided by the government for Malaysian households, which might be one of the contributing factors for insignificant increments in the mean caries.

Lacking oral health awareness is another possible risk factor for caries and obesity. These oral health habits include the frequency of tooth brushing and dental visits, which affect the prevalence of dental caries and gingivitis among children and adolescents. It is suggested in the literature that, apart from variation of weight among participants, positive oral health behavior plays an influential role in participants’ caries experiences [18]. It can be observed in this study—regardless of body weight—that the majority of participants reported “good” in the perception of health and oral health, and practiced tooth brushing twice daily, and the changes in the burden of oral diseases were found not statistically significant (*p* > 0.05).

Study results highlighted the severity of dental caries and gingival bleeding in the Malaysian adolescent population. An increased trend in the prevalence of dental caries was observed with more than half of the participants, regardless of body mass index presented with at least two teeth affected, at the age of 16 years old. The severity of caries was high in the OW/OB group of the SiC subgroup, with at least five teeth affected, with no significant association between dental caries and body mass index. In line with the Malaysia National Oral Health Survey of School Children (NOHSS) 2007 report, about 60% of the Malaysian adolescents, aged 16 year old, were diagnosed with dental caries [19]. This showed that dental caries was a common oral health problem in the population, regardless of body mass index and age. Results from this study also suggested that the association between obesity and dental caries was more complex and a simple explanation based on unhealthy dietary habits and sugar consumption were inadequate. Less than 10% of the OW/OB group from the total study population and in the SiC subgroup presented with healthy gingiva. It is in accordance with the prevalence of healthy periodontium in Malaysian schoolchildren, reported in theNational Health and Morbidity survey (NHMS) 2017 findings [20].

The most prevalent periodontal disease in adolescents was found to be plaque induced gingivitis. Although gingivitis is an irreversible disease in nature, obese adolescents, even without concomitant metabolic syndrome, are more likely to be diagnosed with gingivitis and to develop periodontal disease earlier compared to healthy normal weight adolescents [13]. Therefore, based on the findings of this study, the OW/OB participants, especially in the SiC subgroup, was at a higher risk of being diagnosed with periodontitis in the future. In view of this, further studies should be carried out to further understand the impact of obesity on periodontal disease in healthy adolescents.

The DMFT index was employed to determine the prevalence of dental caries. Despite its limitation of overestimation of dental caries, it has been reliably used in research studies. Recent systematic review and meta analysis also supported the used of DMFT index to standardize the definition of dental caries in the population for further studies[21]. Moreover, the assessment of caries severity at various stages might be suitable to be used in addition to the DMFT index in the study for OW/OB children and adolescents. An earlier study recommended the use of ICDAS criteria as a strong predictor of lesion progression, due to the slow progression of dentinal caries compared to initiation of enamel caries noted from radiographic examination [22]. Observation of differences and development of caries through intraoral examination in obese adolescents can be done through a longer duration of study.

A review also reported no standardization in the diagnostic criteria within the published studies of caries and obesity [23]. It was suggested that, without a gold standard in research methods, it might affect the results and potential findings of the relationship between caries and obesities. 

Interestingly, this study reported low impact in the oral health related quality of life, which was consistent with good self-rated perception of oral health and health among participants regardless of their BMI and severity of oral diseases. In contrast, a study by Banu et al. reported high impact on the OHRQoL [24]. It might be related to the subjective measure by young adolescents regarding their body weight and oral diseases, which indirectly may have influence in their quality of life.

The psychological discomfort was the most affected domain among OW/OB participants with a high burden of dental caries. This is an important finding, which supported other evidence-based studies on mental health, showing that obese adolescents have higher incidence of depression, anxiety, low self-esteem, and poor quality of life [25]. The possible explanation would be victimization by peers, parents, or siblings who shape the belief of having an ideal body weight, defined as beautiful and good for one’s health and oral health. Although OW/OB presented with good perception of health and oral health with low impact on quality of life, it affected the psychological discomfort domain the most. Thus, it is important for policymakers to strengthen health promotion among adolescents, which, in the long run, may help to prevent diseases and health problems, such as obesity, dental caries, and periodontal diseases.

A selection bias was identified as one of the limitations in our study. The cluster random sampling was used to determine samples. This allowed schools located near the Faculty of Dentistry University Malaya to be selected. University Malaya is located in a well-developed area in the Klang Valley. Hence, all participants basically originated from an urban area, which had higher access to healthcare services, with a better education system. Thus, it will be important to have a more diversified environment to determine population impact measures. Although this is a longitudinal study, the duration for the follow-up review was insufficient for the causal relationship between the burden of oral diseases, body mass index, and impact on quality of life to be established. Another follow-up study may be needed to explore further relationships after the peak of growth spurt is achieved. This is because the effect of hormonal changes during puberty varies from one person to another, which may have led to the negative findings from this study.

## 5. Conclusions

In conclusion, this two-year follow-up study suggested that obesity status did not influence the burden of oral disease and OHRQoL. This statement agrees with systematic reviews that report inconclusive and conflicting evidence on the association between obesity and dental caries [20]. However, this study offers insights into changes in oral disease burden and OHRQoL among adolescents. Study limitations highlighted by this study could be discussed further and improved for future studies.

## Figures and Tables

**Table 1 children-08-00435-t001:** Background information of participants at baseline.

Characteristics	Baseline *n* (%)		Follow-Up *n* (%)		*p* Value	Cohen’s Effect Size
	OW/OB *n* = 195	NW *n* = 202	OW/OB *n* = 122	NW *n* = 116		
Gender						
Male	124 (31.2)	118 (29.7)	73 (30.7)	73 (30.7)		*0.03* ^1^
Female	71 (17.9)	84 (21.2)	49 (20.6)	43 (18.1)	*0.81*	
Ethnicity						
Malays	117 (29.5)	105 (26.4)	72 (30.3)	68 (28.6)	*0.05* *	*0.05* ^1^
Non-Malays	78 (19.7)	97 (24.5)	50 (21.0)	48 (20.2)		
Mother level of education						
High	57 (14.4)	67 (16.9)	32 (13.4)	35 (14.7)	*0.79*	*0.02* ^1^
Low	138 (34.8)	135 (34.1)	90 (37.8)	81 (34.0)		
BMI (kg/m^2^) (Mean) (SD)	28.5 (4.01)	19.2 (1.84)	29.6 (4.30)	19.6 (1.67)		*0.02* ^1^

OW = overweight, OB = obese, NW = normal weight. SD = standard deviation, *p* values were calculated using chi square tests, *p* < 0.05 significance values. ^1^ Effects size based on Cohen’s r: 0.2 for small; 0.5 for medium; 0.8 for large effect.

**Table 2 children-08-00435-t002:** Oral hygiene practice and perception on health and oral health in obese, normal weight, and SiC groups at baseline and follow-up.

	Baseline *n* (%)			Follow-Up *n* (%)		
	OW/OB *n* = 122	NW *n* = 116	SIC *n* = 79	OW/OB *n* = 122	NW *n* = 116	SIC *n* = 79
Frequency of tooth brushing	X^2^ = 0.26, *p* = 0.87			X^2^ = 0.49, *p* = 0.77		
2x/day	89 (37.0)	95 (39.9)	64 (81.0)	91 (38.4)	91 (38.4)	65 (82.3)
≤1x/day	33 (14.2)	21 (8.8)	15 (19.0)	31 (12.6)	25 (10.5)	13 (16.5)
Used fluoride toothpaste	X^2^ = 2.28, *p* = 0.13			X^2^ = 0.26, *p* = 0.87		
Yes	120 (50.4)	110 (46.2)	76 (96.2)	78 (35.5)	69 (29.5)	55 (69.6)
No	2 (0.8)	6 (2.5)	3 (3.8)	14 (5.9)	15 (6.3)	24 (30.4)
Used mouth rinse	X^2^ = 0.00, *p* = 0.94			X^2^ = 0.49, *p* = 0.77		
Yes	120 (50.4)	110 (46.2)	21 (26.6)	25 (10.5)	22 (9.3)	15 (19.0)
No	2 (0.8)	6 (2.5)	58 (73.4)	97 (40.9)	93 (39.2)	64 (81.0)
Perception of health	X^2^ = 5.46, *p* = 0.17			X^2^ = 2.98, *p* = 0.08		
Poor	43 (18.1)	25 (10.5)	24 (30.4)	61 (26.1)	46 (19.7)	32 (40.5)
Good	79 (33.2)	91 (38.2)	55 (69.6)	58 (24.8)	69 (29.5)	46 (58.2)
Perception of oral health	X^2^ = 0.00, *p* = 0.97			X^2^ = 3.13, *p* = 0.77		
Poor	45 (18.9)	43 (18.1)	33 (41.8)	65 (27.9)	50 (21.5)	36 (45.6)
Good	77 (32.4)	73 (30.7)	46 (58.2)	53 (22.7)	65 (27.9)	41 (51.9)
Visited dentist for the past 1 year	X^2^ = 1.14, *p* = 0.56			X^2^ = 0.56, *p* = 0.45		
Yes	16 (6.7)	26 (10.9)	12 (15.2)	17 (7.2)	17 (7.2)	8 (10.1)
No	106 (44.5)	90 (37.8)	67 (84.8)	104 (43.9)	99 (41.8)	71 (89.9)

OW = overweight, OB = obese, NW = normal weight, SIC = significant index caries. X^2^ = chi square statistic, *p* values were calculated using chi square tests with *p* < 0.05 is the significance value.

**Table 3 children-08-00435-t003:** Oral health status in obese and normal weight groups of total participants and SiC group at baseline and follow-up.

Total Participants, *n* = 238	Baseline * M ± SD		Follow-Up ** M ± SD		*p* Value
	OW/OB	NW	OW/OB	NW	
Caries experience	1.80 (3.16)	1.84 (2.62)	2.37 (3.30)	2.47 (3.05)	0.91 *, 0.79 **
Based on teeth					
Decayed (D)	1.66 (3.10)	1.53 (2.35)	2.11 (3.16)	2.07 (2.70)	0.71 *, 0.92 **
Missing (M)	0	0.03 (0.18)	0.01 (0.09)	0.06 (0.24)	0.03 *, 0.02 **
Filled (F)	0.13 (0.48)	0.27 (0.92)	0.25 (0.72)	0.34 (1.10)	0.57 *, 0.45 **
Sound dentition *n* (%)	73 (59.8)	60 (51.7)	56 (45.9)	42 (36.2)	
Periodontal status *n* (%)					
BPE screening					
Code 0	8 (6.6)	6 (5.2)	7 (6.0)	10 (8.2)	
Code 1	20 (16.4)	15 (12.9)	49 (42.2)	39 (32.0)	
Code 2	94 (77.0)	95 (81.9)	60 (51.7)	73 (59.8)	
Gingival bleeding *n* (%)					>0.05 ^+^
Yes	118 (96.7)	112 (96.6)	111 (91.0)	106 (91.4)	
No	4 (3.3)	4 (3.4)	11 (9.0)	10 (8.6)	
Significant Caries Index (SiC) *n* =79	5.22 (3.90)	*4.48 (2.70)*	6.49 (3.08)	5.80 (2.73)	0.32 *, 0.28 **
Based on teeth					
Decayed (D)					
Missing (M)	4.89 (4.0)	*3.70 (2.70)*	5.80 (3.46)	4.83 (2.67)	0.11 *, 0.17 **
Filled (F)	0	*0.07 (0.26)*	0.03 (0.16)	0.12 (0.33)	0.10 *, 0.12 **
	0.32 (0.75)	*0.71 (1.42)*	0.68 (1.13)	0.83 (1.68)	0.12 *, 0.63 **
Periodontal status *n* (%)					
BPE screening					
Code 0	0				
Code 1	5 (13.5)	*1 (2.4)*	3 (8.1)	2 (4.8)	
Code 2	32 (86.5)	*5 (11.9)*	19 (51.4)	18 (42.9)	
		*36 (85.7)*	15 (40.5)	22 (52.4)	
Gingival bleeding *n* (%)					
Yes	37 (100)				
No	0	*41 (97.6)*	34 (91.9)	38 (90.5)	>0.05 ^+^
		*1 (2.4)*	3 (8.1)	4 (9.5)	

OW = overweight, OB = obese, NW = normal weight. M = mean, ±SD = standard deviation. *= at baseline, **= at follow up. *p* values were calculated using independent *t*-test, *p* < 0.05 significance values. ^+^ *p* values were calculated using chi square tests, *p* < 0.05 significance values.

**Table 4 children-08-00435-t004:** Oral health related quality of life in obese, normal weight, and SIC groups at baseline and follow-up.

	Baseline * M ± SD			Follow-Up ** M ± SD			*p* Value
	OW/O	NW	SiC ^+^	OW/OB	NW	SiC ^+^	
OHIP severity	9.14 (6.37)	8.63 (5.97)	9.83 (6.68)	6.80 (7.49)	7.52 (6.65)	8.18 (7.47)	0.53 *, 0.75 *^+^, 0.45 **, 0.57 **^+^
OHIP Impact based on items (*n*, %)							
Functional limitation							
(trouble pronouncing words)	0	1 (1.3)	1 (1.3)	1 (0.4)	2 (0.8)	3 (3.8)	
Functional limitation							
(worsened sense of taste)	1 (1.3)	0	1 (1.3)	4 (1.7)	1 (0.4)	8 (10.1)	
Physical pain							
(painful aching in the mouth)	1 (1.3)	1 (1.3)	2 (2.5)	2 (0.8)	2 (0.8)	6 (7.6)	
Physical pain							
(uncomfortable eating)	2 (2.5)	0	2 (2.5)	6 (2.5)	1 (0.4)	6 (7.6)	
Psychological discomfort							
(self-consciousness)	5 (6.3)	8 (10.1)	13 (16.5)	12 (5.0)	16 (6.7)	15 (19.0)	
Psychological discomfort							
(felt tense)	4 (5.1)	2 (2.5)	6 (7.6)	7 (2.9)	9 (3.8)	8 (10.1)	>0.05 ***
Physical disability							
(diet has been unsatisfactory)	2 (2.5)	2 (2.5)	4 (5.1)	5 (2.1)	5 (2.1)	7 (8.9)	
Physical disability							
(meals interrupted)	1 (1.3)	1 (1.3)	2 (2.5)	5 (2.1)	8 (3.4)	6 (7.6)	
Psychological disability							
(difficult to relax)	0	1 (1.3)	4 (5.1)	3 (1.3)	1 (0.4)	3 (3.8)	
Psychological disability							
(been embarrassed)	3 (3.8)	1 (1.3)	1 (1.3)	0	2 (0.8)	4 (5.1)	
Social disability							
(been irritable with others)	8 (10.1)	1 (1.3)	9 (11.4)	17 (7.1)	6 (2.5)	2 (2.5)	
Social disability							
(difficulty doing schoolwork)	0	1 (1.3)	1 (1.3)	0	2 (0.8)	2 (2.5)	
Handicap (felt life is less satisfying)	0	0	0	0	1 (0.4)	2 (2.5)	
Handicap							
(unable to perform usual function)	1 (1.3)	2 (2.5)	3 (3.8)	3 (1.3)	4 (1.7)	5 (6.3)	

OW = overweight, OB = obese, NW = normal weight, SIC = significant index caries. M = mean, ±SD = standard deviation. *, **, ^+^ *p* values were calculated using independent *t*-test, *p* < 0.05 significance values. *** *p* values were calculated using independent *t*-test, *p* < 0.05 significance values. * = at baseline, ** = at follow up.

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
