# Peer review of "Oral Diseases and Quality of Life between Obese and Normal Weight Adolescents: A Two-Year Observational Study"

_children, 2021, doi:10.3390/children8060435_

Round 1

Reviewer 1 Report

Tengku et al. created an interesting work based on many participants' follow-up for two years to assess oral diseases and quality of life between obese and normal weight adolescents. It is a well written and interesting paper that might be of interest to the readers of Children after a few minor remarks.

The paper destroys the existing myth that obese people have a higher oral disease burden than lean people. The essence of the problem is not obesity, but oral hygiene practice, self-consciousness, level of education, and quality of life.

Minor comments

- It would be easier to read the methods section when divided into smaller parts e.g. 

experimental design, anthropometric measurements, dental examination, etc.

-Separate fragment is a section method should be dedicated to statistical analysis. In this section, please determine how you present data. M±SD is unclear. M means mean or median?

- The data in Table 1: OW/OB and NW should be compared with the chi2 test to show that the participants do not differ from each other, of course, except anthropometric measurements. P-value would be desirable. It would be perfect for comparing them at baseline and after following up

-Table 2 - please explain X2

-Line 185: "The mean caries experience among OW/OB and control group in the baseline were 1.80±3.16, 1.84±2.62….." 

- Please determine what data are presented in table3: n (%), mean? median? What is M±SD? The statistical analysis in this table must be improved (Anova, chi2). Chi2 not only in gingival bleeding but also for other categorical variables. The same for table 4.

-Statistical analysis and the presented results must be improved…It is not clear for the readers.

Reviewer 2 Report

The authors performed a two-year study to associate oral diseases related to the quality of life with obese and normal-weight adolescents (14-16 years old). The results from the baseline 397 participants and two-year follow-up 238 participants showed that obesity did not cause influent oral diseases such as dental caries, periodontal diseases, and gingival bleeding. Although this is a negative result, the authors provide a detailed discussion to explain their results. It might be a good reference for a similar investigation.
